# Genetic Dissection of Heat Stress Tolerance in Faba Bean (*Vicia faba* L.) Using GWAS

**DOI:** 10.3390/plants11091108

**Published:** 2022-04-20

**Authors:** Fouad Maalouf, Lynn Abou-Khater, Zayed Babiker, Abdulqader Jighly, Alsamman M. Alsamman, Jinguo Hu, Yu Ma, Nicolas Rispail, Rind Balech, Aladdin Hamweih, Michael Baum, Shiv Kumar

**Affiliations:** 1International Center for Agricultural Research in the Dry Areas (ICARDA), Beirut 1108-2010, Lebanon; lynnaboukhater@hotmail.com (L.A.-K.); r.balech@cgiar.org (R.B.); 2Agricultural Research Cooperation (ARC)-Hudeiba Sudan, Wad Madani 21111, Sudan; z_babiker@yahoo.com; 3Agriculture Victoria, AgriBio, Centre for AgriBiosciences, Bundoora, VIC 3083, Australia; abdulqader.jighly@agriculture.vic.gov.au; 4Agricultural Genetic Engineering Research Institute, Cairo P.O. Box 12619, Egypt; smahmoud@ageri.sci.eg; 5USDA-ARS Plant Germplasm Introduction & Testing Research Unit, Pullman, WA 99163, USA; jinguo.hu@usda.ov; 6Department of Horticulture, Washington State University, Pullman, WA 99164, USA; yu.ma@wsu.edu; 7Institute for Sustainable Agriculture, CSIC, 14004 Córdoba, Spain; nrispail@ias.csic.es; 8ICARDA, Cairo P.O. Box 12619, Egypt; a.hamweih@cgiar.org; 9Biodiversity and Integrated Gene Management Program, ICARDA, 10106 Rabat, Morocco; m.baum@cgiar.org (M.B.); sk.agrawal@cgiar.org (S.K.)

**Keywords:** faba bean, heat stress, sequencing, single nucleotide polymorphism, genome-wide association study, single trait, multi-trait

## Abstract

Heat waves are expected to become more frequent and intense, which will impact faba bean cultivation globally. Conventional breeding methods are effective but take considerable time to achieve breeding goals, and, therefore, the identification of molecular markers associated with key genes controlling heat tolerance can facilitate and accelerate efficient variety development. We phenotyped 134 accessions in six open field experiments during summer seasons at Terbol, Lebanon, at Hudeiba, Sudan, and at Central Ferry, WA, USA from 2015 to 2018. These accessions were genotyped using genotyping by sequencing (GBS), and 10,794 high quality single nucleotide polymorphisms (SNPs) were discovered. These accessions were clustered in one diverse large group, although several discrete groups may exist surrounding it. Fifteen lines belonging to different botanical groups were identified as tolerant to heat. SNPs associated with heat tolerance using single-trait (ST) and multi-trait (MT) genome-wide association studies (GWASs) showed 9 and 11 significant associations, respectively. Through the annotation of the discovered significant SNPs, we found that SNPs from transcription factor helix–loop–helix bHLH143-like S-adenosylmethionine carrier, putative pentatricopeptide repeat-containing protein At5g08310, protein NLP8-like, and photosystem II reaction center PSB28 proteins are associated with heat tolerance.

## 1. Introduction

Faba bean (*Vicia faba* L.) is the fourth most important cool season legume crop grown globally in diverse cropping systems and environments [1]. It plays a valuable role in increasing food production and enhancing sustainable farming methods due to its ability to fix nitrogen and improve soil structure [2,3]. Faba bean is currently cultivated on 2.65 million ha worldwide, with a total dry grains production of 5.6 Mt [4]. This crop is used as human food and animal feed, mainly for pigs, horses, poultry, and pigeons, in many countries [5] because it is rich in protein and essential amino acids [6] such as arginine, lysine, and leucine. It also has high iron (22 to 78 mg/kg) and zinc (45–61 mg/kg) contents [7]. On the other hand, several constraints, such as biotic and abiotic stresses, significantly affect its productivity. Heat stress is one of the major constraints affecting faba bean in heat prone environments.

Due to climate change, heat waves (periods of abnormally hot weather lasting from days to weeks) are expected to become more intense and frequent, and the average global temperature is also expected to increase by 1.5 °C, as reported by the assessments of the Intergovernmental Panel on Climate Change (IPCC) (https://www.ipcc.ch/sr15/, accessed on 5 July 2021). This will have a severe impact on crop production in many regions, especially in the Mediterranean region and Nile Valley countries such as Ethiopia, Egypt, and Sudan, where faba bean is grown extensively as a major food legume crop. Moreover, the projected climate change will affect the stability of faba bean cultivars, which highlights the need to develop heat tolerant faba bean cultivars with stable floral and reproductive development [8]. Heat stress can significantly reduce faba bean yield [9], especially when the temperature rises above 30 °C [10]. Heat stress during flowering reduces pollen viability, which reduces the pod set [11]. The effects of heat stress on pollen development have also been reported in many other legumes such as chickpea (*Cicer arietinum*) [11], common bean (*Phaseolus vulgaris*) [12], cowpea (*Vigna unguiculata*) [13], groundnut (*Arachis hypogaea*) [14], soybean (*Glycime max*) [15], and lentil (*Lens culinaris*) [16].

The field selection method is used in heat prone environments to identify sources of heat tolerance and results in the development of heat tolerant varieties [7]. These conventional breeding tools are effective but time consuming. Although recent advances in the next generation sequencing technologies (NGS) have facilitated the generation of large volumes of sequences [17,18,19], the large genome size of faba bean (~13 Gb) has so far limited the development of comprehensive genomic information [20,21,22]. However, NGS can be easily used to discover single nucleotide polymorphisms (SNPs) associated with key traits through genome-wide association studies (GWASs) [17,22,23]. GWASs utilize the concept of linkage disequilibrium (LD) in diverse populations to detect SNPs associated with different traits [24]. One of the key factors that affects the power of GWASs is the size and structure of the population used for the analysis [25]. The population can be computationally increased using multi-variate (environments/traits) genome-wide association study (M-GWAS) models if the evaluation trial is not large enough [26].

Breeding new genotypes with tolerance/avoidance to projected stresses is important when it comes to thoroughly quantifying the crop’s response to heat stress [27]. So far, efforts made to explore the physiological and molecular basis of heat tolerance have been mainly focused on vegetative growth [28] and reproductive stages [11,29]. SNPs associated with traits of interest in faba bean have been reported for some biotic stresses such as Ascochyta blight resistance [30,31] and broomrape resistance [31], for quality traits including low vicine-convicine content [32], and for herbicide tolerance [33]. However, no comprehensive study has been conducted to detect molecular markers associated with heat tolerance.

The major objective of the present research was to identify markers associated with heat tolerance in a diverse faba bean collection. The secondary objective was to study the relationship between grain yield and different physiological and agronomical traits in diverse heat prone environments.

## 2. Results

### 2.1. Phenological and Yield Traits 

Combined variance analysis using residual maximum likelihood (REML) and spatial models for each trial are presented in Table 1. Significant differences were observed among genotypes and among environments for all evaluated traits. The genotype × environment (G × E) interaction (Table 2) were significant for all traits except for grain yield, indicating that, contrary to the order of other traits, the order of yield performance of the tested accessions remained the same across the environments. Spatial analysis for each season–location combination indicated a wide variability among genotypes for days to flowering (DFLR) and days to maturity (DMAT) at Terbol and Hudeiba, from 2015 to 2017. Significant differences were observed among genotypes for grain yield (*p* < 0.001 at all locations/seasons in 2016 and 2017 and *p* < 0.05 at Terbol in 2015). Significant differences were also found among genotypes for the number of seeds per plant and hundred seed weight in all environments. Plant height had significant variation among genotypes at all locations except for Hudeiba in 2017. Means, ranges, and standard errors scored for all traits and locations are presented in Table 3. The average grain yield varied from 1.3 g at Hudeiba station to 13.9 g at the experimental station in Central Ferry, indicating wide variability across locations. Large variability between locations were also reported for other traits. The hundred seed weight showed consistently lower values under heat stress at Hudeiba, Sudan.

### 2.2. Physiological Traits

Combined variance analysis (Table 1) showed significant differences among accessions for PG and CL. However, no significant difference was observed for canopy temperature (CT) in both seasons at Hudeiba station, Sudan. Significant differences were detected among environments (*p* < 0.001), and accession X environment were significant for PG and CL. Means and ranges for CL and PG are presented in Table 3. The average PG varied from 8.87% in 2015 to 28.4% at Terbol station, while CL varied from 40.3 to 41.2% at Hudeiba.

### 2.3. Relationship between Traits

Since CT did not vary significantly among accessions, it was excluded from the correlation and regressions analysis. Correlations and stepwise regression analysis using mean estimates are presented in Table 4. GY was significantly and positively correlated with NPP (*p* < 0.001), PG and PLHT (*p* < 0.01), and NSP (*p* < 0.05). HSW was significantly and positively correlated with PLHT (*p* < 0.001) but negatively correlated with PG (*p* < 0.001). CL was highly correlated with PLHT, but it was not correlated with GYP under heat stress. Therefore, CL appeared inefficient for selection for yield in a heat prone environment, despite it being an effective indicator of plant biomass.

DFLR was positively correlated with DMAT and negatively correlated with HSW and PLHT. Early flowering lines with late maturity showed higher seeds sizes than late flowering lines. The regression coefficient (Table 4) indicated that GYP had moderate correlation with NPP (α = 0.31), PG (α = 0.1), and HSW (α = 0.01). These results indicated that PG and NPP are effective parameters to select for heat tolerance in faba bean. A total of 15 lines from different subspecies (Table 5) were selected based on grain yield, pollen germination, and plant height, of which eight were *equina* types, two belong to *major* types, three to *minor* types and two to *paucijuga*.

### 2.4. Population Structure 

A principal component analysis (PCA) is presented in Figure 1. The results showed that the tested accessions were clustered in one large group, although three discrete groups of genotypes may exist surrounding it. Labelling each faba bean genotype according to its subspecies or geographic origin does not discriminate any clear group, suggesting that the assembled faba bean germplasm is composed of a single divergent population (Figure 1).

### 2.5. Genome-Wide Association Analysis

Analysis of 39 traits across six environments using ST-GWAS revealed 68 SNP–trait associations for 25 traits, while the remaining 14 traits had no associations. Figure 2 depicts the QQ plots that were used to compare observed vs. expected values to demonstrate how well the GWAS model performed in identifying trait-associated SNPs. All traits except CL exhibited associations in at least one environment. The highest number of associations were detected for NSP, NPP, and GYP in Terbol summer 2016 with 9, 7, and 6 associations, respectively. Out of these 68 associations, only 14 were significant following Bonferroni correction at *p* < 4.6 × 10^−6^ while the remaining 54 associations were suggestive (Appendix A). Nine SNPs showed associations with multiple traits/environments, of which the SNP SCONTIG60075_82 showed the highest number of four associations with GYP, NPP, and NSP in Hudeiba 2017 as well as NSP in Hubeiba 2016. The SNP SNODE_39949_LENGTH_82_COV_1.024390_37 showed associations with NPP, GYP, and NSP in Hudeiba 2017. The remaining eight SNPs showed associations with two traits, with five being associated with NSP and NPP in Terbol summer 2016, two associated with GYP and NSP in Hudeiba 2016, and the remaining one associated with NPP and GYP in Terbol summer 2015 (Appendix A). 

The MT-GWAS model which fit each of the 10 studied traits across all environments in a single analysis detected a total of 56 SNP-trait associations, of which 11 were significant when considering Bonferroni correction (Table 6, Figure 2). These associations were clustered in 50 QTL based on linkage disequilibrium (LD) analysis. NPP showed 10 QTL, followed by NSP, which had nine QTL, while DMAT and CL showed one QTL each. Each of the following traits, GYP, HSW, and PG, had seven QTL, while PLHT showed four QTL and CT and DFLR had only two QTL each (Appendix A). Like the ST-GWAS analysis, the SNPs SNODE_39949_LENGTH_82_COV_1.024390_37 and SCONTIG60075_82 showed association with three traits, GYP, NPP, and NSP. The SNP SNODE_40333_LENGTH_77_COV_34.987015_87 was associated with GYP and NPP, while the SNP SCONTIG124089_41 was associated with both NPP and NSP.

### 2.6. Candidate Gene Annotation

Ten identified SNPs located within diverse functional genes are presented in Table 7 and Table 8. These SNP markers were associated with PG, HSW, NSP, and GYP using MT-GWAS. The annotated SNP markers associated with PG were: (1) Contig82855 located within one of two probable polygalacturonase genes related to lupin (*Lupinus angustifolius*), jequirity bean (*Abrus precatorius*), and the pseudomolecule Pd05 gene of sweet almond (Prunus dulcis); (2) NODE_38942_length_69_cov_118.768112 located within one of seven uncharacterized LOC100810394 genes related to soybean, jequirity bean, chickpea, and barrel clover (*Medicago truncatula*); (3) NODE_6662_length_69_cov_474.000000 located within one of three genes of transcription factor-basic helix–loop–helix (TF-bHLH143) related to chickpea, barrel clover, and lupin; (4) NODE_7398_length_62_cov_214.516129 located within six genes of S-adenosylmethionine carrier 1 related to chickpea and barrel clover; and (5) NODE_7979_length_116_cov_512.344849 located within one of seven putative pentatricopeptide repeat-containing protein At5g08310 genes related to chickpea and barrel clover.

The annotated SNP markers associated with HSW were: (1) Contig16540 located within one of the two uncharacterized genes (LOC109813943) related to pigeon pea (*Cajanus cajan*), two uncharacterized genes (LOC101492966) of chickpea, as well as to the complete sequenced gene clone mth2-173c1 of barrel clover; (2) NODE_8714_length_71_cov_9.901408 located within cilia- and flagella-associated protein 251-like (LOC112012620) related to cork oak (*Quercus suber*); and (3) NODE_9908_length_67_cov_43.89552 located within one of three genes (mth2-176a22, mth2-18p3 map mtgsp_014c01 and mth2-64j6) related to barrel clover.

In addition, SNP Contig82391 and NODE_14795_length_67_cov_68.791046 were respectively associated with NSP located within one of two proteins, NLP8-like (LOC101496898) genes of chickpea, and with GYP located within the LOC25500962 gene of barrel clover.

Finally, Contig60075, which was a major SNP marker identified by ST-GWAS, was located within photosystem II reaction center PSB28 protein. This marker is associated with DFLR, DMAT SNP, NPP, and GYP.

## 3. Discussion

Heat stress during the reproductive phase significantly affects faba bean yield. The identification of a heat tolerant faba bean is therefore required to assure appropriate yield in heat prone environments. Yield and yield component-related traits are highly affected by heat stress in faba bean [8,10]. The results of our field experiments confirmed the results of earlier studies that heat stress negatively affects faba bean yield and yield components such as pollen germination, pod number, seed number, and grain yield. Grain yield was found to be positively associated with pod and seed numbers. Heat stress therefore damages faba bean during the flowering period, when the viability of pollen is critical for successful reproduction [8]. Moreover, our experiments were conducted under heat prone and well-watered conditions, and therefore canopy temperature was not associated with grain yield, as reported in wheat [34]. Moreover, in agreement with the results obtained under water stress conditions in faba bean [35], chlorophyll content and plant height were not associated with grain yield.

To date, conventional breeding methods have identified only limited heat tolerant faba bean cultivars [36]. These methods are relatively slow in achieving significant progress. The development of genetic markers associated with important traits under heat stress in faba bean is essential to improve selection accuracy and make selections in early generations. Such markers are effective for selecting adapted and tolerant accessions. Few published resources have used SNP genotyping to provide a useful genetic map that could be applied in faba bean breeding [17]. In the present study, GBS technology was used to genotype 134 faba bean accessions that belong to four different botanical varieties originated from either the collection of landraces from 42 countries or from ICARDA breeding program. Our GBS pipeline produced 10,794 SNPs on a diverse faba bean germplasm. Recently, SNP genotyping analysis was used to generate a collection of 1819 gene-based SNP markers in three recombinant inbred line (RIL) populations to enable the design of a high-density consensus genetic map in faba bean [22]. The high number of SNPs produced in the present study may be due to the high heterogeneity, as we used diverse genotypes. A parental population with a common ancestor produces a low amount of genetic variability and therefore a low number of SNPs [37].

Despite the large genetic variability observed in the faba bean germplasm, only one large heterogeneous group was observed by PCA (Figure 1), confirming the lack of genetic or sterility barriers between different subspecies [38]. This contrasts with the previous ADMIXTURE analysis that reported the presence of two ancestral subpopulations within the germplasm [33]. While the germplasm used in both studies largely overlap, the previous one included 52 additional accessions, all belonging to the first subpopulation, while the shared accessions all belonged to the second subpopulation. 

A limited number of studies have investigated the genetic control of different traits in faba bean through biparental populations, multi-parental populations, or GWAS [22,30,33,39], but none of these targeted QTL associated with heat tolerance. Thus, the QTL identified for heat tolerance in the present study would be very useful in faba bean breeding programs. Moreover, the QTL detected with the MT-GWAS model are supposed to be stable across environments, giving them a broader practical impact in breeding [26,40] 

Heat stress has harmful effects on plant metabolism and impacts crop growth and development throughout vegetative and reproductive phases [41]. Plants exposed to heat stress generate reactive oxygen species (ROS), antioxidant substances, plant hormones, and other secondary metabolites [41]. Therefore, heat stress response pathways rely heavily on transcription factor-mediated gene expression regulatory networks. Annotated genes related to SNPs under heat stress are the factors regulating heat tolerance in faba bean. As an example, SNPs associated with PG under heat stress were located within transcription factor (TF) bHLH143-like, probable polygalacturonate inhibiting protein (PGIP), S-adenosylmethionine carrier 1 (SAMC), and putative pentatricopeptide repeat-containing protein At5g08310 (RCP). Therefore, these genes may have crucial roles in controlling pollen germination (PG) under heat stress. Similar results showed that TF and PPRCP genes had crucial roles in the resistance of *Benincasa hispida* to high temperature [42]. In addition, SMAC was considered as an important precursor of polyamine and ethylene and leads to stress tolerance in plants [43]. 

Psb28 is an extrinsic protein of photosystem II (PSII) that is found in photosynthetic organisms ranging from cyanobacteria to higher plants. Several studies have revealed that Psb28 is involved in the recovery of PSII under high temperature stress [44]. Additionally, the correlation between gene expression of Psb28 and plant response to heat stress was detected in several plant species including *Populus tomentosa* and perennial ryegrass [45,46]

The SNPs marker Contig82391 located with *Cicer arietinum* protein NLP8-like (LOC101496898) was found to be associated with the number of seeds per plant under heat stress. NLP8 was described in Arabidopsis as an essential protein for nitrate-promoted seed germination [47].

## 4. Materials and Methods

### 4.1. Genetic Materials

The present study involved a set of 134 faba bean accessions comprising 16 breeding lines developed at ICARDA and 118 landraces originating from 42 countries. The sub-species of the germplasm set was characterized as 42 *major*, 17 *minor*, 66 *equina*, and 9 *paucijuga*. This germplasm set represents wide diversity, as it represents the 1000 accessions previously assessed by Maalouf et al [20].

### 4.2. Experimental Designs 

Six experiments were conducted in an incomplete block design with two replicates at three locations between 2015 and 2017 (Table 9). Experiments details are described as follows: (1)Three experiments were conducted during the summer season at the ICARDA Terbol station (35.98 N, 33.88 E, 890 m above sea level). The summer season at the Terbol station, Lebanon runs from June to October and is characterized by hot and dry weather, with temperatures above 35 °C during the flowering and pod set time (Figure 3a). A 50 mm irrigation was provided to the crop every seven days using drip irrigations to ensure enough moisture for crop growth. The soil in Terbol station is a deep and rich clay loam soil.(2)Two experiments were conducted during the winter season at the ARC Hudeiba Research Station in Sudan (17.56° N, 33.93° E, and 350 m above sea level) in high terrace soil (Almatra) during the 2015/16 and 2016/17 seasons. The winter season at Hudeiba runs from November to March and is characterized by hot and dry weather with daytime temperatures above 32 °C (Figure 3b). Flood irrigation was provided at 10 day intervals to ensure enough moisture in the soil.(3)One experiment was conducted at Central Ferry Research Farm, USA (Central Ferry, Pullman, WA; 46°43′52′′ N, 117°39′52′′ W) during the spring season from April to August 2017. The Central Ferry location has a Chard silt loam soil (coarse-loamy, mixed, super active, mesic Calcic Haploxerolls). The season is characterized by warm weather during the reproductive and terminal crop cycle (Figure 3b). Using subsurface drip irrigation, 10 mm of water irrigation were provided daily.

The plot size in all six experiments was 2 m in length with two rows 45 cm apart. Soil fertilization of NPK 15-15-15 at 250 kg ha^−1^ was added; weeds were controlled by the pre-emergence application of pendimethalin at 1200 g a.i./ha followed by manual weeding; insects and fungal diseases were controlled regularly to eliminate any additional stress on plants other than the heat stress.

### 4.3. Phenotyping for Heat Stress 

The population was phenotyped for days to 50% flowering (DFLR) and days to 90% maturity (DMAT) on plot basis and plant height (PLHT), number of pods per plant (NPP), number of seeds per plant (NSP), and grain yield per plant (GYP) as an average of three random plants per plot. Hundred seed weight (HSW) was recorded as the weight of a hundred random seeds per plot. In addition, observations on chlorophyll content and canopy temperature were also recorded at Hudeiba research station during both seasons. 

Chlorophyll content (Cl): The chlorophyll content of three random leaves per plot was measured during morning hours at the end of the flowering period using a chlorophyll meter (SPAD-502 Minolta (Spectrum Technologies Inc., Plainfield, IL, USA). The SPAD 502 Plus Chlorophyll Meter instantly measures chlorophyll contents to assess the effect of heat stress on vegetative growth and its relation to yield and other agronomic traits. 

Canopy temperature depression (CT): This was measured during vegetative and grain filling stages (CTvg and CTgf, respectively) using a portable infrared thermometer (Mikron M90 Series, Mikron Infrared Instrument Co., Inc., Oakland, NJ, USA) (CT). 

Pollen Germination (PG): The pollen germination test was conducted during three consecutive seasons from 2015 to 2017 at Terbol station. Fresh pollens were randomly collected from 15 opened flowers of each plot when the day temperature exceeded 32 °C. The collected pollens were then kept in microfuge tubes in iceboxes for a few hours prior to the germination test. The pollen germination test was conducted in two replicates per plot following the method described earlier [48]. A medium containing 0.1 g/L boric acid (H3BO3), 0.3 g/L calcium nitrate (Ca (NO3)2), 100 g/L sucrose, and 10 g/L agar (Sigma, Aldrich-Germany) was prepared and autoclaved at 121 °C before being poured into Petri dishes (20 mL/dish of 9 cm diameter). The pH was adjusted to 6.3. The freshly collected pollen grains were dusted and dispersed on the germination surface using a needle and then incubated at 37 °C for 16 h before the germination was terminated by adding a drop of acetocarmine to the medium. Pollen grains were counted as germinated if the elongation of pollen tubes were at least equal to the diameter of the grain. The germination rate was determined by counting the germinated pollens per hundred observed pollen grains in each replicate under a Cole-Parmer DX53056101 Stereozoom Microscope, with a built-in 2.0 mega pixel digital camera, 110–220 VAC.

### 4.4. DNA Extraction and Genome by Sequencing Analysis 

Details relating to the DNA extraction, GBS protocol, and data analysis were previously described in Abou Khater et al. [33]. In brief, genomic DNA was extracted using the Qiagen Plant DNA Preparation Kit, while the two restriction enzymes used to generate the GBS libraries, PstI and MspI, were prepared with 48 barcode adapters with a 4–9 bp sequence [49] and produced approximately four million single reads (100 bp) per genotype using the Illumina HiSeq 2500 (San Diego, CA, USA) using standard Illumina protocols and kits which produce high-output paired-end 100 bp reads (Montreal, QC, Canada). The TASSEL-GBS 3.0 bioinformatics analysis package was used to process raw sequence files in FASTQ format using the first steps of the TASSEL-GBS pipeline with its default parameters. The TASSEL-GBS workflow was used to trim reads, remove barcodes, and generate a single tag count file for each sample. The raw sequence files in FASTQ format were assembled and converted into FASTA format, from which a faba bean reference genome was constructed using NCBI and pulsedb genomic and transcriptomic sequences, as well as our newly assembled sequences. This step should result in a unified reference genome that covers more transcriptionally active regions of the faba genome. GBS tags were aligned to the generated reference genome using Bowtie2 V2.2.4 [50] with the very-sensitive-local option. The generated BAM files were used in the TASSEL-GBS pipeline for SNP marker calling. SNPs with >85% call rate and >5% minor allele frequency were used in subsequent analyses.

### 4.5. Statistical Analysis of Phenotyping Data 

Variance analysis using the spatial model for all traits was applied using GenStat V19 with the automatic incomplete block design modules [51]. The replicates and block within replicates were fitted as random variables and accessions were considered as fixed variables. In addition to variance parameters, the output of the above model displays Wald statistics, the predicted mean value of each accession, and the standard error. Differences among accessions and environments were examined using the multi-environment trials analysis (META), in which accessions were fitted as fixed variables while environments and the accession × environment interaction (G × E) were fitted as random variables. The META analysis was conducted using the method of residual maximum likelihood (REML).

Correlation analysis and step-wise regression analysis were conducted to examine the relationship between traits and to define traits that have an influence on grain yield in faba bean under heat prone conditions. The combined mean estimates were obtained and used for correlations and step-wise multiple regression analyses. The selected heat tolerant accessions are supposed to have higher yield and higher values for traits influencing yield under stress conditions.

### 4.6. Population Structure and Genome-Wide Association Analysis

PCA was estimated using TASSEL 5 [52] on the 10,794 SNPs. The PCA vectors were then plotted in R [53] with the ggplot2 R package [54]. Faba bean accessions were labelled with different colors and shapes according to their geographic origin and subspecies, respectively. 

Two GWAS models were used in the present study, the single-trait (ST) model, in which each environment is fitted independently, and the multi-trait (MT) model, in which all environments are jointly fitted in a single analysis. GEMMA software was used to fit both models with its default parameters and was also used to calculate the genomic relatedness matrix (GRM) following the VanRaden (2008) method [55]. Significant threshold was defined with Bonferroni correction at *p* < 0.05 and all SNP-trait associations that had *p* < 1 × 10^−4^ were reported as suggestive associations. To determine if different significant SNPs for each analysis were associated with the same quantitative trait locus (QTL) or different QTL, the linkage disequilibrium (LD) among them was calculated following the Weir method [56]. The sequences flanking associated SNPs were blasted against the NCBI database to annotate potential candidate genes underlying the causal variants.

## 5. Conclusions

Heat stress is one of the major constraints affecting faba bean in many production regions, especially when waves of high temperatures coincide with the reproductive phase. It also limits the crops expansion in new geographic regions where high temperatures are common during the growing season. By improving heat tolerance in faba bean, the crop can be expanded to new niches in sub-Saharan Africa and therefore contribute to the improvement of soil quality. The identified heat tolerant lines can be used to develop cultivars adapted to heat prone areas and potentially contribute to enhanced food security in countries such as Sudan, Egypt, and Ethiopia. Considering the importance of heat tolerance in faba bean, it is imperative to breed elite cultivars that feature this trait. However, field selection is very costly and time consuming, and this is reflected by low genetic gains over the years. The integration of genomic selection and marker-assisted selection will improve selection accuracy and intensity and will shorten the breeding cycle when selecting at early generations. In the present study, we identified SNP markers associated with agronomic and physiological traits under heat prone conditions using single- and multi-trait association. These SNP markers located within functional genes expressed under heat stress will facilitate and fasten the development of heat tolerant cultivars and can be used to achieve the introgression of heat tolerant genes into a desired agronomic background.

## Figures and Tables

**Figure 1 plants-11-01108-f001:**
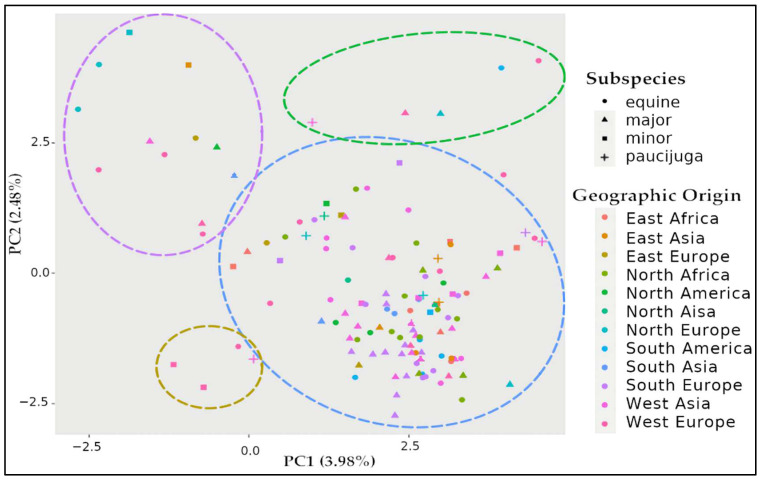
Scatterplot of principal component analysis scores of components 1 and 2 of faba bean panel. PCA was estimated in TASSEL and represented in R. Genotypes are identified by color according to their geographic origin and shape according to their subspecies. Potential clusters of genotypes are indicated by colored circles.

**Figure 2 plants-11-01108-f002:**
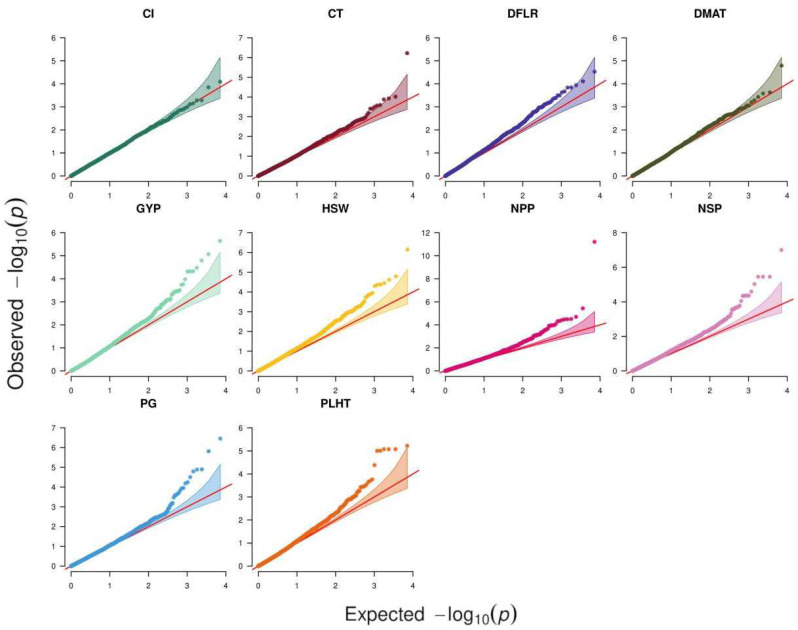
Q–Q plots of the chi2 statistics obtained from processing traits and genomic variations of the studied faba accessions. GY: grain yield per plant; DFLR: days to flowering after sowing; DMAT: days to maturity after sowing; PLHT: plant height; NPP: number of pods per plant; NSP: number of seeds per plant; HSW: hundred seed weight in g; PG: pollen germination under heat stress conditions with temperature above 32-degree Celsius; Cl: chlorophyll content.

**Figure 3 plants-11-01108-f003:**
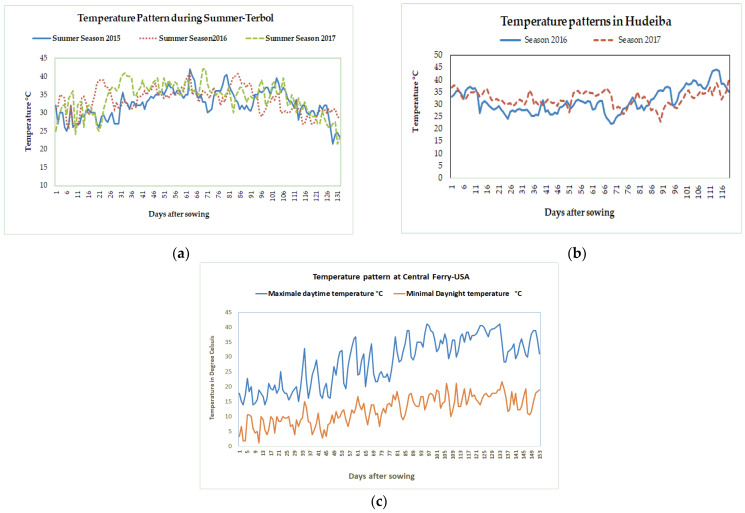
Variation in maximal temperature during the growing seasons from 2015–2017 at Terbol, Lebanon (**a**), Hudeiba, Sudan (**b**), and Central Ferry, WA, USA (**c**).

**Table 1 plants-11-01108-t001:** Spatial model analysis performed for detecting significance differences in genotypic variation of phenotypic and physiologic traits, expressed as *p*-value.

	Terbol, Lebanon			Hudeiba, Sudan		Pullman, WA, USA
	2015	2016	2017	2016	2017	2017
GY	0.017	<0.001	<0.001	<0.001	<0.001	<0.001
PLHT	<0.001	<0.001	<0.001	<0.001	0.335	<0.001
DFLR	<0.001	<0.001	<0.001	<0.001	<0.001	NA
DMAT	NA	<0.001	0.004	<0.001	0.003	NA
NPP	0.003	<0.001	<0.001	<0.001	<0.001	NA
NSP	0.004	<0.001	<0.001	0.003	<0.001	<0.001
CL	NA	NA	NA	0.004	0.009	NA
PG	0.015	<0.001	<0.05	NA	NA	NA
HSW	0.049	<0.001	<0.001	<0.001	<0.001	<0.001

GY: grain yield per plant in g; DFLR: days to flowering after sowing; DMAT: days to maturity after sowing; PLHT: plant height in cm; NPP: number of pods per plant; NSP: number of seeds per plant; HSW: hundred seed weight in g; PG: pollen germination under heat stress conditions with temperature above 32-degree Celsius; Cl: chlorophyll content. NA: not applicable.

**Table 2 plants-11-01108-t002:** Combined analysis using residual maximum likelihood (REML) for detecting significance differences in genotypic variation of phenotypic and physiologic traits, expressed as *p*-value.

	Genotypes (G)	Environment (E) *	G × E
DF	133	5	665
GY	<0.001	<0.001	<0.398
PLHT	<0.001	<0.001	<0.001
DFLR	<0.001	<0.001	<0.001
DMAT	<0.001	<0.001	<0.001
NPP	<0.001	<0.001	<0.001
NSP	<0.001	<0.001	<0.001
CL	<0.001	0.118	<0.001
PG	<0.001	<0.001	<0.001
HSW	<0.001	<0.001	<0.001

DF: degree of freedom; GY: grain yield per plant in g; DFLR: days to flowering after sowing; DMAT: days to maturity after sowing; PLHT: plant height in cm; NPP: number of pods per plant; NSP: number of seeds per plant; HSW: hundred seed weight in g; PG: pollen germination under heat stress conditions with temperature above; 32-degree Celsius; Cl: chlorophyll content. NA: not applicable. * Df = 5 for GY, NSP and HSW, Df = 4 for DFLR, DF = 3 for DMAT, DF = 2 for PG and DF = 1 for CL.

**Table 3 plants-11-01108-t003:** Means and ranges for phenotypic and physiological traits recorded in different locations and seasons.

		GY	DFLR	DMAT	PLHT	NPP	NSP	HSW	PG	Cl
Year	Summer Terbol, Lebanon									
2015	Range	0.0–55	36–102	NA	16.7–8	0.0–47.6	0.0–30.5	13–156	0.0–50.4	NA
	Mean	9.0	51.85	NA	60.89	7.63	10.15	68.95	8.87	NA
2016	Range	0.0–29	34–65	90–152	18–80	NA	0–31	20–200	0.0–73	NA
	Mean	1.4	46	94.5	54.29	NA	2.07	80.43	15.8	NA
2017	Range	0.9–12	34.7–74.5	111–121	20–92.3	0–22.6	0–163.8	3–103	0.0–120	NA
	Mean	13.7	49.7	117.4	65.9	7.3	32.3	23.3	28.4	NA
	Hudeiba, Sudan									
2016	Range	0.0–11	27.5–91	91–114	21.3–64.9	0.0–21.1	0.0–35.5	9.9–88	NA	28–65
	Mean	1.26	50.68	103	48.8	1.62	2.72	47.68	NA	41.2
2017	Range	0.0–12	NA	63–117	14.1–66.9	0.0–22.14	0.0–20.5	NA	NA	17–50
	Mean	1.3	NA	107	42.86	2.22	1.56	NA	NA	40.3
	Central Ferry, WA, USA									
2017	Range	1.3–29	NA	NA	33.9–81.8	NA	3–16	20–194	NA	NA
	Mean	13.96	NA	NA	59.36	NA	14.01	98.5	NA	NA

GY: grain yield per plant in g; DFLR: days to flowering after sowing; DMAT: days to maturity after sowing; PLHT: plant height in cm; NPP: number of pods per plant; NSP: number of seeds per plant; HSW: hundred seed weight in g; PG: pollen germination under heat stress conditions with temperature above 32-degree Celsius; Cl: chlorophyll content. NA: not applicable.

**Table 4 plants-11-01108-t004:** Correlation analysis between mean phenotyping estimates for different phenotyping and physiological traits and regression equation between grain yield and associated traits.

	DFLR	DMAT	GYP	HSW	NPP	NSP	PLHT	PG
DMAT	**0.26 ****	-						
GYP	0.12	−0.02	-					
HSW	**−0.31 *****	−0.13	0.11	-				
NPP	−0.02	−0.07	**0.37 *****	**−0.19 ***	-			
NPP	0.07	0.10	**0.21 ***	0.14	**−0.29 ****	-		
PLHT	**−0.40 *****	−0.06	**0.25 ****	**0.36 *****	−0.14	0.17	-	
PG	0.12	0.00	**0.29 ****	**−0.32 *****	**0.32 ****	−0.04	−0.06	
CL	−0.05	−0.07	0.07	0.0	−0.03	−0.09	**0.25*****	0.03
GY=+2.5+0.035 HSW+0.31 NPP+0.1 PG (DF=123;p<0.001)								

* *p* < 0.05; ** *p* < 0.01; *** *p* < 0.001. GY: grain yield per plant in g; DFLR: days to flowering after sowing; DMAT: days to maturity after sowing; PLHT: plant height in cm; NPP: number of pods per plant; NSP: number of seeds per plant; HSW: hundred seed weight in g; PG: pollen germination under heat stress conditions with temperature above 32-degree Celsius; Cl: chlorophyll content.

**Table 5 plants-11-01108-t005:** Selected accessions tolerant to heat based on grain yield gram per plant (GYP), plant height in cm (PLHT), number of pods per plant (NPP), and pollen germination (PG).

	IG	ORIGIN	PLHT	NPP	GYP	PG
subsp. faba var. *major*	VF420	Afghanistan	53.34	4.85	5.21	20.2
	FB2648	France	61.79	11.65	20.14	29.41
subsp. faba var. *equina* Pers	IG11908	Ethiopia	56.55	8.53	12.01	17.91
	IG11982	Iraq	58.29	11.13	13.01	17.05
	IG12110	Algeria	60.78	8.98	9.87	26.24
	IG13945	Sudan	67.13	9.17	21.7	18.49
	IG99664	ICARDA	63.78	10.67	9.51	44.82
	Vf351	Turkey	58.62	11.27	13.53	25.23
	FB2509	France	60.12	12.2	10.25	57.01
	IG14026	ETH	59.77	7.47	8.25	16.76
subsp. faba var. *minor* pers	IG12659	Ethiopia	48.49	8.74	12.32	18.39
	IG13958	Syria	63.13	9.87	6.34	34.22
	FB1165	Spain	55.87	9.79	11.21	22.61
subsp. *paucijuga* Beck	Vf301	Czech Republic	45.83	11.4	7.68	19.2
	VF626	Unknown	10.758	**54.5**	6.901	27.44
Mean of tested populations			7.2	34.5	5.15	14
Standard error			7.9	5.15	5.1	14.5

**Table 6 plants-11-01108-t006:** SNPs detected in multi-trait genome-wide association study (MT-GWAS) with significant *p*-values. AF: allele frequency of allele1.

Trait	QTLID	SNP	Allele1	Allele0	AF	P
NPP	NPP_8	SNODE_40333_LENGTH_77_COV_34.987015_87	T	A	0.18	3.70 × 10^−6^
	NPP_9	SNODE_559376_LENGTH_95_COV_1.252632_60	A	T	0.45	6.20 × 10^−12^
NSP	NSP_4	SCONTIG82391_71	T	G	0.09	3.5 × 10^−6^
	NSP_4	SCONTIG82391_72	C	T	0.09	3.5 × 10^−6^
	NSP_4	SCONTIG82391_73	A	T	0.09	3.5 × 10^−6^
	NSP_5	SNODE_11884_LENGTH_82_COV_596.182922_61	A	G	0.09	1.0 × 10^−7^
HSW	HSW_6	SNODE_9908_LENGTH_67_COV_43.895523_45	A	G	0.17	7.1 × 10^−7^
GYP	GYP_3	SCONTIG72702_49	A	G	0.5	2.3 × 10^−6^
CT	CT_2	SCONTIG50196_81	C	T	0.06	6.0 × 10^−7^
PG	PG_1	SCONTIG82855_50	A	G	0.09	1.50 × 10^−6^
	PG_6	SNODE_7398_LENGTH_62_COV_214.516129_82	G	A	0.16	3.50 × 10^−7^

**Table 7 plants-11-01108-t007:** Gene annotations of different identified SNPs markers associated with pollen germination.

Markers	Gene Names
Contig82855	Lupinus angustifolius probable polygalacturonase (LOC109360706), mRNA
	Abrus precatorius probable polygalacturonase (LOC113847809), mRNA
	Prunus dulcis DNA, pseudomolecule Pd05
NODE_38942_length_69_cov_118.768112	Medicago truncatula uncharacterized LOC11423568, mRNA
	Glycine soja uncharacterized LOC114380151, mRNA
	Abrus precatorius uncharacterized LOC113852670, transcript variant X3, mRNA
	Glycine max uncharacterized LOC100810394, mRNA
	Medicago truncatula clone mth2-17i21, complete sequence
	Cicer arietinum uncharacterized LOC101512103, mRNA
	Abrus precatorius uncharacterized LOC113852670, transcript variant X2, mRNA
NODE_6662_length_69_cov_474.000000	Cicer arietinum transcription factor bHLH143-like (LOC101493666), mRNA
	Lupinus angustifolius transcription factor bHLH143-like (LOC109349271), mRNA
	Medicago truncatula transcription factor bHLH143 (LOC11430352), mRNA
NODE_7398_length_62_cov_214.516129	Cicer arietinum S-adenosylmethionine carrier 1, chloroplastic/mitochondrial (LOC101510252), transcript variant X1, mRNA
	Medicago truncatula S-adenosylmethionine carrier 1, chloroplastic/mitochondrial (LOC11420332), transcript variant X3, misc_RNA
	Cicer arietinum S-adenosylmethionine carrier 1, chloroplastic/mitochondrial (LOC101510252), transcript variant X3, mRNA
	Medicago truncatula S-adenosylmethionine carrier 1, chloroplastic/mitochondrial (LOC11420332), transcript variant X1, mRNA
	Cicer arietinum S-adenosylmethionine carrier 1, chloroplastic/mitochondrial (LOC101510252), transcript variant X2, mRNA
	Medicago truncatula S-adenosylmethionine carrier 1, chloroplastic/mitochondrial (LOC11420332), transcript variant X2, mRNA
NODE_7979_length_116_cov_512.344849	Medicago truncatula putative pentatricopeptide repeat-containing protein At5g08310, mitochondrial (LOC11440721), transcript variant X1, mRNA
	Medicago truncatula putative pentatricopeptide repeat-containing protein At5g08310, mitochondrial (LOC11440721), transcript variant X2, mRNA
	Lupinus angustifolius putative pentatricopeptide repeat-containing protein At5g08310, mitochondrial (LOC109335950), mRNA
	Medicago truncatula clone mth2-123f23, complete sequence
	Medicago truncatula putative pentatricopeptide repeat-containing protein At5g08310, mitochondrial (LOC11440721), transcript variant X3, mRNA
	Cicer arietinum putative pentatricopeptide repeat-containing protein At5g08310, mitochondrial (LOC113783927), mRNA
	Medicago truncatula putative pentatricopeptide repeat-containing protein At5g08310, mitochondrial (LOC11440721), transcript variant X4, mRNA

**Table 8 plants-11-01108-t008:** Gene annotations of different identified SNPs markers associated with phenological and yield traits.

Markers	Gene Names
Hundred seed weight (HSW)	
Contig16540	Cajanus cajan uncharacterized LOC109813943, transcript variant X2, mRNA
	Cajanus cajan uncharacterized LOC109813943, transcript variant X1, mRNA
	Medicago truncatula uncharacterized LOC11425609, mRNA
	Medicago truncatula clone mth2-173c1, complete sequence
	Cicer arietinum uncharacterized LOC101492966, transcript variant X1, mRNA
	Cicer arietinum uncharacterized LOC101492966, transcript variant X2, mRNA
NODE_8714_length_71_cov_9.901408	Quercus suber cilia- and flagella-associated protein 251-like (LOC112012620), partial mRNA
NODE_9908_length_67_cov_43.895523	Medicago truncatula clone mth2-176a22, complete sequence
	Medicago truncatula clone mth2-18p3 map mtgsp_014c01, complete sequence
	Medicago truncatula clone mth2-64j6, complete sequence
Number of seeds per plant (NSP)	
Contig82391	Cicer arietinum protein NLP8-like (LOC101496898), transcript variant X2, mRNA
	Cicer arietinum protein NLP8-like (LOC101496898), transcript variant X1, mRNA
Grain yield per plant (GYP)	
NODE_14795_length_67_cov_68.791046	Medicago truncatula uncharacterized LOC25500962, mRNA
	GYP, NSP, NPP, DFLR, DMAT
Contig60075 *	photosystem II reaction center PSB28 protein

* ST-GWAS method was found in Hudeiba during 2016 and 2017 season.

**Table 9 plants-11-01108-t009:** Details of experiments conducted in different locations.

	Locations	Period of Cropping	Irrigation Pattern	Day Time Max T	Nighttime Max T °C
2015	Terbol	June–October	Drip irrigation 50 mm/week	35 °C	19 °C
2016	Terbol	June–October	Drip irrigation basis 50 mm/week	35 °C	19 °C
	Hudeiba	November–March	Flood irrigation every 10 days	36 °C	19 °C
2017	Terbol	June–October	Drip irrigation basis 50 mm/week	36 °C	20 °C
	Hudeiba	November–March	Flood irrigation every 10 days	40 °C	20 °C
	Pullman	April–August	Drip irrigation 10 mm/day	>40 °C	21 °C

## Data Availability

The datasets generated and analyzed during the current study are available from the corresponding authors on request.

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
