# Peer review of "Genetic Dissection of Heat Stress Tolerance in Faba Bean (Vicia faba L.) Using GWAS"

_plants, 2022, doi:10.3390/plants11091108_

Round 1

Reviewer 1 Report

Dear Authors

The present study aimed to identify sources of tolerance to heat stress and to identify markers associated with heat tolerance in a diverse faba bean collection. The study was planned nicely and present in well organised manner. Faba bean is a very important crop and may be a potencial material as an alternative to Soyabean. Soyabean is promoting monoculture practices and this may be incorpoarted in itroduction here (in the manuscript) as well. Second thing is, Faba bean has some NNCs (namely vicine and convicine) which are major limitations for this crop. This point may also be adreesed in the manuscript.

Thank you

Author Response

We are very thankful for the evaluation made by reviewer one. We strongly agree that faba bean has potential to replace soybean, in cool season environments and also anti-nutritional components require special attention to make the substitution of soybean by faba bean in animal feeding happening. 

Reviewer 2 Report

This study aimed to identify markers associated with heat stress tolerance and correlate different physiological and agronomical traits with the yield. Generally, the paper is well written and deserves to be published. There are some suggestions:

L 82 Please clarify the main objectives of the paper

Figure 1 Add the clear mark for different clusters

It is necessary to discuss the results from points 2.2 and 2.3.

Author Response

  • We agree with reviewer regarding the aim of this study and the main objective in L82 was modified accordingly in the text.
  • Figure modified as requested, and we made slight changes in the text:  ".... tested accessions were clustered in one large group although several discrete groups may exist surrounding it. Labelling each faba bean genotype according to its subspecies or geographic origin does not discriminate any clear group suggesting that the assembled faba bean germplasm is composed of a single divergent population.." (line 152-157)
  • The discussion of results of point 2.2 and 2.3 was made from line 242 to 248 in the discussion part.

Reviewer 3 Report

The authors have done a great job, the structure of the experiment, the methods used correspond to the conclusions of the authors and speak of the reliability of the data obtained.
There are some remarks about the design, especially the tables, when they are torn into three pages, they become almost unreadable. I recommend that you think about this point, and improve the quality of your presentation. Also lines 9 and 14 - both under the number "4", but different affiliations.

Author Response

Thank you very much for the time used to review the paper. As per your suggestions, long tables have been split into small one based and marked in red in the text

In addition, the number “4” in line 14 was corrected and replaced by number 9.
